# LEARNING KEY STEPS TO ATTACK
# DEEP REINFORCEMENT LEARNING AGENTS

## ABSTRACT

Deep reinforcement learning agents are known to be vulnerable to adversarial attacks. In particular, recent studies have shown that attacking a few key steps is effective for decreasing the agent's cumulative reward. However, all existing attacking methods find those key steps with human-designed heuristics, and it is not clear how more effective key steps can be identified. This paper introduces a novel reinforcement learning framework that learns key steps through interacting with the agent. The proposed framework does not require any human heuristics nor knowledge, and can be flexibly coupled with any white-box or black-box adversarial attack scenarios. Experiments on benchmark Atari games across different scenarios demonstrate that the proposed framework is comparable and could be superior to existing methods for learning key steps.

## 1 INTRODUCTION

Reinforcement learning (RL) is a framework for sequential decision problems, where an agent interacts with an unknown environment and tries to maximize the total reward it receives. With the rapid development of deep learning, RL agents parametrized by neural networks, usually referred to as deep RL agents, are able to learn complex policies from raw inputs (Mnih et al., 2015). Recently, deep RL agents have shown great success across various domains, such as achieving superhuman performance on games (Mnih et al., 2015; Silver et al., 2016; 2018), completing complex robotic tasks (Levine et al., 2016), optimizing patient treatments (Escandell-Montero et al., 2014; Raghu et al., 2017), and developing autonomous driving skills (Pan et al., 2017; Isele et al., 2018).

However, if we wish to deploy these RL agents into security-critical applications, we must take their reliability into consideration. It is discovered that neural network classifiers may make errors on deliberately crafted inputs, known as adversarial examples (Biggio et al., 2013; Szegedy et al., 2013; Behzadan & Munir, 2017a). Deep RL is no exception: Many works have demonstrated that deep RL agents are also vulnerable when attacked by adversarial examples (Behzadan & Munir, 2017a; Huang et al., 2017), raising serious concerns on the reliability of these agents for security-critical applications. For example, there may be severe consequences if an RL-agent-driven autonomous vehicle is compromised by adversarial examples fed from a malicious attacker.

Given the sequential nature of RL, it is not necessary to attack at every time step to degrade the agent's performance significantly (Lin et al., 2017; Kos & Song, 2017). The reason is that not all decisions made by the agent are equally important. Some decisions may be critical to the agent, such as those for long-term planning or immediate reward gathering; some other decisions may not have much effect on the environment nor the rewards, and thus attacking those steps would not affects the agent's performance. The critical steps will be named *key steps* in this work, and attacking these steps decreases the cumulative reward that the agent could gather.

Figure 1 illustrates a key step and a non-key step in Atari Pong, where the main difference between them is whether the ball has already passed through the agent. If the attacker is aware of this difference, the attacker can concentrate on attacking the key steps. By attacking fewer steps, the attacker reduces the costs of generating, injecting, and hiding the adversarial examples. Therefore, an intelligent attacker should prefer attacking key steps over non-key steps.

A natural question arises: How does an attacker identify the key steps to attack? As an initial attempt to address this question, Lin et al. (2017) formulate the problem from an optimization perspective,

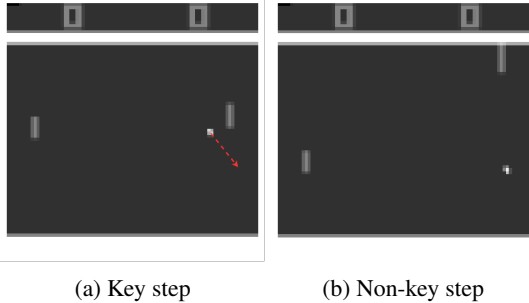

(a) Key step          (b) Non-key step

Figure 1: An example of the key step and non-key step in Atari Pong. (a) The ball is moving in the direction indicated by the red arrow. If the agent does not move down at this step or some earlier steps, the ball would pass through, making the agent lose a point. (b) The ball has already passed through the agent. Any decision at this step does not save the situation and does not affect the environment much.

and propose a heuristic that attacks the agent when it strongly prefers one action. Concurrently, Kos & Song (2017) propose another heuristic that attacks when the action appears rewarding to the agent. However, since these heuristics are designed based on human knowledge, it is unclear whether there exist more effective key steps. In this work, we study the possibilities of finding more effective key steps, and of finding them automatically without human knowledge. We confirm both possibilities by introducing an RL framework where the attacker *learns* the key steps *from scratch* through interacting with the agent.

In particular, we study the *key-step identification problem* proposed by Lin et al. (2017). We first make a crucial observation that this constrained optimization problem can be converted into another form that matches the objective of RL. Then, we design a corresponding RL environment to train the attacker. Our contributions are summarized as follows:

- We formulate the key-step identification problem in an unconstrained form, and introduce an RL framework that solves it directly without incorporating human knowledge. The proposed RL framework is independent of how adversarial examples are generated, and thus could be combined with white-box or black-box attacks.

- We justify our learning framework by giving a toy example where those human-designed heuristics fail to identify the key steps.

- To test the effectiveness of our framework, we conduct experiments on four benchmark Atari games. The results suggest that the attacker trained by our framework learns effective key steps and has the potential to outperform prior methods.

## 2 BACKGROUND AND RELATED WORK

### 2.1 REINFORCEMENT LEARNING BACKGROUND

We begin with an introduction to the Markov Decision Process (MDP) (Sutton & Barto, 2018). For a set $\mathbb{S}$, we use $\Delta(\mathbb{S})$ to denote the set of probability distributions over $\mathbb{S}$. Consider an agent interacting with an environment: At each step $t$, the agent picks an action $a_t$ based on a current state $s_t$, and then the environment returns a reward $r_t$ and a next state $s_{t+1}$. We formalize this interaction as an MDP $\mathcal{M}$, which is a 5-tuple $(\mathbb{S}, \mathbb{A}, P, R, \gamma)$, where $\mathbb{S}$ is a finite set of states, $\mathbb{A}$ is a finite set of actions, $P \colon \mathbb{S} \times \mathbb{A} \to \Delta(\mathbb{S})$ defines the transition dynamics from $s_t$ to $s_{t+1}$, $R \colon \mathbb{S} \times \mathbb{A} \to \Delta(\mathbb{R})$ defines the reward distribution, and $\gamma \in [0, 1]$ is a discount factor. For convenience, we use $r \colon (s_t, a_t) \mapsto \mathbb{E}_{r_t \sim R(s_t, a_t)}[r_t]$ to denote the mean reward function. We specify an agent by its (stochastic) policy $\pi \colon \mathbb{S} \to \Delta(\mathbb{A})$, and define its Q-value function as

$$Q^\pi(s_t, a_t) = \mathbb{E}\left[\sum_{k=0}^{\infty} \gamma^k r(s_{t+k}, a_{t+k})\right],$$

where $s_i \sim P(s_{i-1}, a_{i-1})$ and $a_i \sim \pi(s_i)$ for all $i > t$. The agent's goal is to find a policy that maximizes the expected cumulative reward, or expected return, defined by $Q^\pi(s_1, \pi(s_1))$, starting from an initial state $s_1$. For finite-horizon MDPs, we use $T$ to denote the maximum number of steps, and define an episode to be a sample of $(s_1, a_1, r_1, \ldots, s_T, a_T, r_T, s_{T+1})$.

To learn the optimal policy, deep Q-network (DQN, Mnih et al. 2015) first approximates the Q-value function by a neural network that takes a state as input and outputs the Q-values for all actions. The policy is then induced by picking the action with the largest Q-value at each state. Given a transition step $(s_t, a_t, r_t, s_{t+1})$, the network $f_\theta$ is trained by minimizing the square loss

$$\mathcal{L}_\theta = \left( \left( r_t + \gamma \max_a f_\theta(s_{t+1})_a \right) - f_\theta(s_t)_{a_t} \right)^2,$$

where we use $f_\theta(s)_a$ to denote the $a$-th entry of the Q-values $f_\theta(s)$.

## 2.2 ADVERSARIAL ATTACK BACKGROUND

In the context of adversarial attacks, an attacker tries to craft adversarial examples that the target model would misclassify. In gradient-based adversarial attacks, the attacker generates adversarial examples by adding small perturbations to data examples, such that the perturbations are imperceptible to human eyes. Take the fast gradient sign method (FGSM) by Goodfellow et al. (2014) for example. Given a target model $f_\theta$ parametrized by $\theta$ and an image–label pair $(x, y)$, the FGSM computes the perturbation as

$$\eta = \epsilon \, \mathrm{sign}(\nabla_x J(\theta, x, y)),$$

where $\epsilon$ is a scaling factor that controls the norm of the perturbation, and $J(\theta, x, y)$ is an objective function that depends on the attacker's goal. In untargeted attacks, the attacker aims to minimize the target model's classification accuracy. Thus, the objective $J(\theta, x, y)$ is substituted with $D_{\mathrm{KL}}(e^{(y)} \| f_\theta(x))$, the Kullback-Leibler divergence between the label's one-hot encoding $e^{(y)}$ and the predicted probabilities $f_\theta(x)$. However, this type of white-box attacks requires full knowledge of the network parameter $\theta$, which may be hidden in practical tasks.

In contrast, black-box attacks assume no access to $\theta$ and study other conditions, such as the ability to query model outputs (Chen et al., 2017). One black-box attack of particular interest is the substitute model approach (Papernot et al., 2016a; 2017), where the attacker computes perturbation using a substitute model that is trained to perform the same task as the target model. This approach is based on the finding that adversarial examples are transferable; that is, if an adversarial example fools a model, it usually fools another models trained to perform the same task, regardless of the network architecture (Szegedy et al., 2013). Consequently, once the attacker gains access to the training set, black-box attacks reduce to white-box attacks with a lower attack success rate.

## 2.3 ADVERSARIAL ATTACK ON DEEP RL AGENTS

As opposed to traditional adversarial attacks, attacking an RL agent is a unique and challenging task. When we attack a classifier, we typically aim to minimize the classification accuracy or to maximize the probability that the classifier predicts some given class. When we attack an RL agent, however, we usually do not care about the individual actions that the agent picks. In this paper, we choose to minimize the agent's cumulative reward (Huang et al., 2017; Lin et al., 2017) as the attacker's goal. Other possible goals include luring the agent into a designated state (Lin et al., 2017), or misguiding the agent to optimize an adversarial reward (Tretschk et al., 2018).

Previous works have investigated the effects of white-box and black-box attacks against RL agents on the Atari benchmark. Given full knowledge of the target agent's network parameters, an attacker can craft adversarial examples to fool the target agent into picking wrong actions (Behzadan & Munir, 2017a; Huang et al., 2017; Kos & Song, 2017; Mandlekar et al., 2017; Pattanaik et al., 2018; Hussenot et al., 2019) using the FGSM. Other white-box methods such as the Jacobian saliency map algorithm (Papernot et al., 2016b) and the attack proposed by Carlini & Wagner (2017) are also effective against RL agents (Behzadan & Munir, 2017a; Lin et al., 2017). Even if the attacker cannot access the target agent's network parameters, adversarial examples can still be crafted with a substitute agent that is trained in the same environment as the target agent (Behzadan & Munir, 2017a; Huang et al., 2017), similar to the substitute model approach.

Another line of studies tries to improve the RL agent's robustness with adversarial attacks. Several works improve the agent's robustness to visual perturbations through adversarial training (Kos & Song, 2017; Mandlekar et al., 2017; Behzadan & Munir, 2017b; Pattanaik et al., 2018), which is a defense technique that adds adversarial examples into training sets (Goodfellow et al., 2014). On the other hand, Pinto et al. (2017) propose to improve the agent's robustess to environment changes by training the agent and an adversary in an alternating procedure, where the attacks are operated on the environment dynamics. Tessler et al. (2019) study the agent's robustness to action-space perturbations in continuous control domains, and achieve better performance even in the absence of the adversary.

# 3 LEARNING KEY STEPS TO ATTACK

## 3.1 THE KEY-STEP IDENTIFICATION PROBLEM

Suppose that an attacker would like to attack a target agent with policy $\pi$ by adding perturbations to the states. We use $\eta$ to denote an arbitrary attack method that maps from a state to a perturbation. Let $B$ be the budget, the maximum number of attacks permitted in one episode. The key-step identification problem (Lin et al., 2017) can be formulated as follows:

$$
\min_{b_1,\ldots,b_T \in \{0,1\}} \quad \mathbb{E}\left[\sum_{t=1}^{T} r(s_t, a_t)\right]
$$

$$
\text{s.t.} \quad a_t \sim \pi(s_t + b_t \eta(s_t)), \ s_{t+1} \sim P(s_t, a_t), \ \forall 1 \le t \le T, \tag{1}
$$

$$
\sum_{t=1}^{T} b_t \le B.
$$

In this problem. the attacker aims to minimize the expected return of the target agent under the budget constraint. The binary variable $b_t$ indicates whether the perturbation $\eta(s_t)$ is added to the state $s_t$, and $t$ is a key step found by the attacker if $b_t = 1$.

Problem (1) is a difficult combinatorial problem with exponentially large search space. Each decision of the attacker changes the subsequent steps. Moreover, the transition dynamics and reward distribution may be random. Even if the attacker manages to search the whole space by brute force, the collected episodes are merely samples. The attacker still needs to estimate the expected return.

Due to these difficulties, previous works (Lin et al., 2017; Kos & Song, 2017) design simple heuristics to find the key steps. Suppose that the target agent is parametrized by a Q-network $f_\theta$.[1] Kos & Song (2017) observe that steps with large Q-value are usually followed by large immediate reward, and hypothesize that an attack is effective at steps with large Q-value. They propose to set $b_t = 1$ if the maximum Q-value $\max_a f_\theta(s_t)_a$ is larger than a given threshold. On the other hand, Lin et al. (2017) propose to add perturbations when the agent is confident about its action. They use the softmax function to convert the Q-values into a probability distribution (let $\pi = \text{softmax} \circ f_\theta$), and set $b_t = 1$ if the probability gap $\max_a \pi(s_t)_a - \min_a \pi(s_t)_a$ is larger than a given threshold.

## 3.2 AN RL FRAMEWORK FOR LEARNING THE KEY STEPS

To address these challenges, we design an RL framework that directly solves the key-step identification problem. Based on the Lagrange relaxation technique (Bertsekas, 1997), we first replace the hard budget constraint with a soft penalty term:

$$
\min_{b_1,\ldots,b_T \in \{0,1\}} \quad \mathbb{E}\left[\sum_{t=1}^{T} r(s_t, a_t)\right] + \lambda \sum_{t=1}^{T} b_t
$$

$$
\text{s.t.} \quad a_t \sim \pi(s_t + b_t \eta(s_t)), \ s_{t+1} \sim P(s_t, a_t), \ \forall 1 \le t \le T, \tag{2}
$$

where $\lambda$ is a parameter that controls the penalty for each attack. Now Problem (2) becomes a sequential decision problem without additional constraints. We then propose to solve this problem through

---

[1]These heuristics can be applied to attack other value-based or policy-based agents in a similar way.

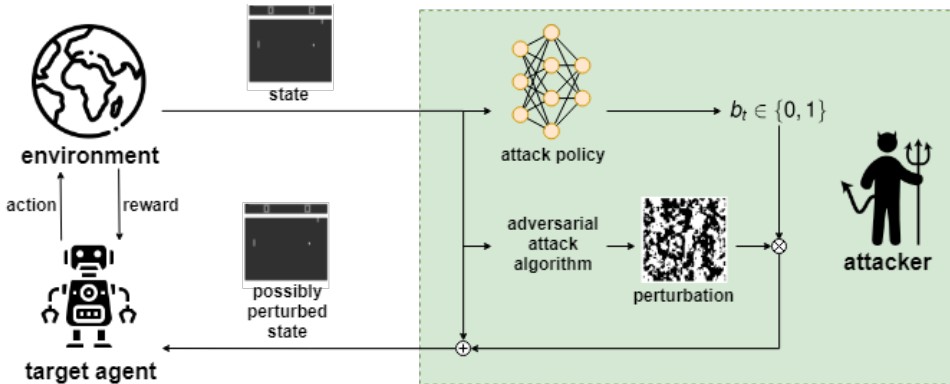

Figure 2: Interactions among the environment, the target agent, and the attacker. If the attack policy outputs "1", the attacker would intercept the state from the environment and inject the perturbed state to the target agent. Different from prior methods, our method parametrizes the attack policy by a neural network and trains it by an RL framework.

RL by training an *attack policy* that identifies the key steps. Figure 2 illustrates the interactions among the environment, the target agent, and the attacker.

Here we describe our proposed RL framework in details. Suppose that the target agent with policy $\pi$ is trained to maximize the expected return in an MDP $\mathcal{M} = (\mathbb{S}, \mathbb{A}, P, R, \gamma)$. We propose to train the attack policy $\pi'$ in another MDP $\mathcal{M}' = (\mathbb{S}', \mathbb{A}', P', R', \gamma')$, where each component is defined as follows:

- $\mathbb{S}' = \mathbb{S}$, and $\gamma' = \gamma$.

- The attacker's action space $\mathbb{A}' = \{0, 1\}$.

- For all $s \in \mathbb{S}'$, $b \in \mathbb{A}'$, the transition dynamics $P'(s, b) = P(s, a)$, and the reward $R'(s, b) = -R(s, a) - b\lambda$, where $a \sim \pi(s + b\eta(s))$ is the target agent's action.

The new environment $M'$ has the same state space as $M$, but reduces to binary action space with the action "1" representing an attack at that step and "0" otherwise. The attacker's reward is the negative of the target agent's reward, plus a penalty of $\lambda$ for each attack. Therefore, maximizing the expected return in $\mathcal{M}'$ is equivalent to minimizing the objective in Problem (2).

Our proposed RL framework is general. We make no assumptions on the environment or on the target agent. In addition, our RL framework can be freely paired with any RL algorithm and any attack method when training the attack policy. Thus, both white-box and black-box attack scenarios can be considered in this framework.

### 3.3 THE POTENTIAL OF LEARNING THE KEY STEPS

Here we justify our RL framework by giving a toy example where the prior methods fail to identify the key steps. Consider a simple MDP shown in Figure 3. Suppose that we would like to attack an agent with its policy and Q-value function given in the figure. If we set the budget constraint $B = 2$, the most effective attack policy should attack steps 1 and 3. However, both heuristics are unable to find these two key steps.

This toy example highlights the potential drawback of these heuristics. Since the attacker might change the agent's policy in the future steps, the agent's Q-value estimates in the current step could be inaccurate. As a result, heuristics based on these Q-value estimates may not find the most effective key steps. In contrast, our framework trains the attacker from scratch. We do not rely on human-designed heuristics, and thus the learned attacker may be able to identify the most effective key steps in this example.

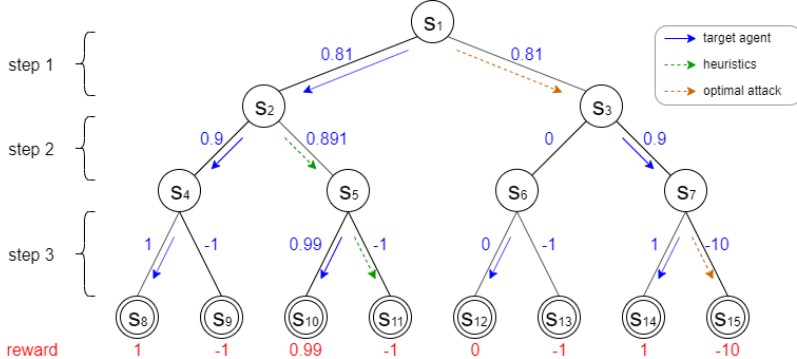

Figure 3: An MDP example where the two heuristics fail. The MDP components are described as follows. A circle represents a state, and a double circle represents a terminal state. At each state, the agent can choose to go left or right, as shown by the lines. Red numbers under the terminal states represent the reward an agent gets if the agent reaches that state. The discount factor $\gamma$ is set to 0.9. Suppose we would like to attack an agent with its estimated Q-values and policy shown by the blue numbers and blue arrows. Let the budget constraint $B = 2$. To minimize the agent's cumulative reward, a smart attacker should attack at steps 1 and 3, guiding the agent into state $s_{15}$. However, the heuristic proposed by Kos & Song (2017) would attack at steps 2 and 3 since the maximum Q-value is larger at those steps. On the other hand, the heuristic proposed by Lin et al. (2017) also attacks steps 2 and 3 since the gap of Q-values is larger at those steps (so the probability gap is larger too). As a result, both heuristics guide the agent into state $s_{11}$, failing to find the best key steps.

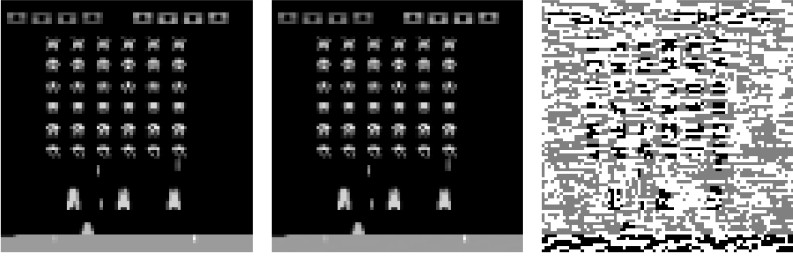

Figure 4: An example of perturbation generated by the FGSM with $\epsilon = 0.01$. *Left:* Original frame. *Middle:* Perturbed frame. *Right:* Added perturbation (rescaled to $[0, 1]$).

## 4 EXPERIMENTS

### 4.1 EXPERIMENTAL SETUP

We evaluate the proposed RL framework on four Atari 2600 games in the Arcade Learning Environment (Bellemare et al., 2013) (Pong, Space Invaders, Seaquest, and Riverraid), which covers a variety of environments. The environment setting and preprocessing match the guidelines suggested by Machado et al. (2018). In particular, we use a sticky-action probability of 0.25 in these environments. For the target agent, we take pretrained DQN agents from Dopamine (Castro et al., 2018), which have the same network architecture as Mnih et al. (2015), and fix them thereafter. We train the attack policy using DQN with this same architecture. It is worth mentioning that we train each attack policy for 10M frames, which is only 1/20 of the frames used to train the target agents. Other hyperparameters are reported in Appendix (Table 2).

Throughout the experiments, we use the untargetted FGSM to generate perturbations due to its computation efficiency. Although we only test our framework in this setting, we expect the framework to generalize to the case of targetted attack, or to other adversarial attack algorithms. We use Foolbox (Rauber et al., 2017) to generate the adversarial examples, with the norm constraint $\epsilon$ set to 0.01. Figure 4 shows an example of the perturbation, which is imperceptible to human and can only be visualized after rescaled. To make the scenario more realistic, we follow Hussenot et al. (2019) and

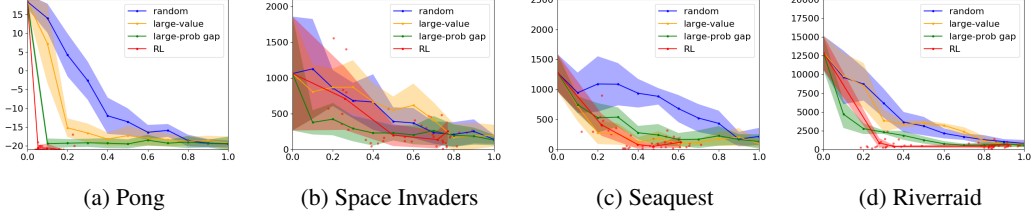

(a) Pong      (b) Space Invaders      (c) Seaquest      (d) Riverraid

Figure 5: Key-step identification with white-box attacks. The vertical axis is the target agent's undiscounted return, and the horizontal axis is the attack ratio (the number of attacks divided by the number of steps in an episode). Under the same attack ratio, the lower the target agent's undiscounted return is, the better the attack method does. All reported scores are averaged over ten testing episodes and shown with one standard deviation. The results of the two heuristics are obtained by setting ten different thresholds on their corresponding criteria. For our method, we train five attack policies using penalty parameters $\lambda \in \{10^1, 10^0, 10^{-1}, 10^{-2}, 10^{-3}\}$, and plot the score for each learned policy, with testing episodes shown by red dots.

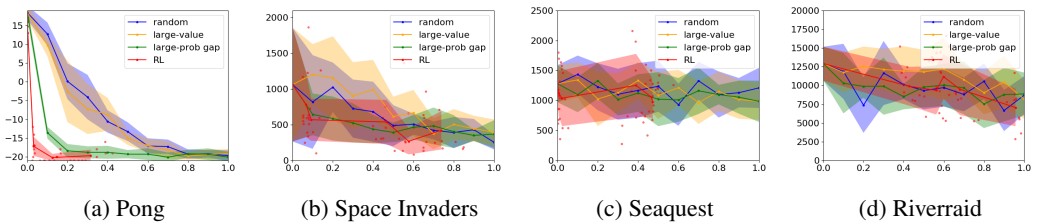

(a) Pong      (b) Space Invaders      (c) Seaquest      (d) Riverraid

Figure 6: Key-step identification with black-box attacks. See Figure 5 for descriptions.

perturb the target agent's observation, rather than the whole state. That is, although the target agent stack the latest four frames as its current state, the attacker is only allowed to add perturbation to the latest frame.

## 4.2 PERFORMANCE COMPARISON OF THE ATTACK POLICY

We compare the attack policy trained by our *RL* framework, to a *random* baseline that attacks uniformly at random, and to two heuristics, *large-value* (Kos & Song, 2017) and *large-prob-gap* (Lin et al., 2017). Figure 5 plots the results in the white-box setting, where the attacker has full knowledge of the target agent's network parameters. The attack policy learned in our RL framework shows comparable performance to human-designed heuristics across different attack ratios consistently, and achieves superior performance in Pong and Riverraid. These results suggest that the key steps are learnable, and that it is possible to identify more effective key steps using our method.

We also test our framework with black-box attacks, where the attacker cannot access the target agent's network parameters. We adopt the substitute model approach, and take the substitute agent from Dopamine, which is a DQN agent trained with a different random seed. The substitute agent is used to compute the perturbations and the statistics needed in the two heuristics. The results are shown in Figure 6. Compared to white-box attacks, black-box attacks are of less success rate, and thus the effect of the attacks is weakened (cf. Figure 5). Still, our method is able to perform comparably to (in Space Invaders, Seaquest, Riverraid) or outperform (in Pong) previous methods.

## 4.3 THE EFFECT OF THE PENALTY PARAMETER

To investigate the penalty parameter $\lambda$, we summarize the performance of the learned attack policies with white-box attacks in Table 1. Results obtained with black-box attacks can be found in Appendix (Table 3). As $\lambda$ decreases, our RL framework tends to learn an attack policy with higher attacker ratio. Training with too large values of $\lambda$ would restrict the attack policy from launching any attacks. Therefore, choosing an appropriate $\lambda$ is important for learning effective key steps. Empirically, we

Table 1: Comparison of different $\lambda$ with white-box attacks. The results are averaged over ten episodes with the standard deviation shown in parentheses.

| | | $\lambda$ | | | | |
|---|---|---|---|---|---|---|
| | | $10^1$ | $10^0$ | $10^{-1}$ | $10^{-2}$ | $10^{-3}$ |
| Pong | return | 16.3 (2.1) | 16.3 (2.1) | -20.5 (0.7) | -20.6 (1.2) | -20.9 (0.3) |
| | attack ratio (%) | 0.0 (0.0) | 0.0 (0.0) | 5.3 (1.0) | 9.7 (4.7) | 15.8 (3.1) |
| Space Invaders | return | 1693.0 (820.0) | 714.0 (438.7) | 201.5 (152.5) | 139.5 (124.6) | 254.0 (147.4) |
| | attack ratio (%) | 0.0 (0.0) | 26.0 (4.9) | 50.0 (9.4) | 71.9 (5.3) | 77.1 (8.1) |
| Seaquest | return | 1288.0 (483.0) | 414.0 (183.3) | 84.0 (72.6) | 122.0 (94.0) | 54.0 (34.7) |
| | attack ratio (%) | 0.0 (0.0) | 21.8 (4.6) | 39.8 (7.7) | 61.1 (5.2) | 47.7 (5.2) |
| Riverraid | return | 888.0 (636.1) | 407.0 (127.3) | 546.0 (113.6) | 616.0 (328.7) | 464.0 (295.9) |
| | attack ratio (%) | 27.8 (5.8) | 35.3 (7.0) | 83.6 (5.0) | 92.2 (5.6) | 82.5 (2.6) |

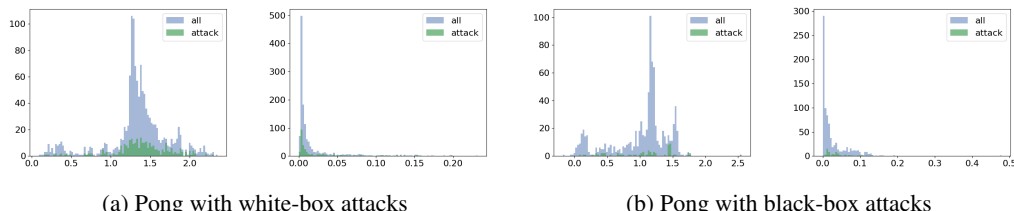

(a) Pong with white-box attacks       (b) Pong with black-box attacks

Figure 7: Behavior comparison of our method (with $\lambda = 10^{-2}$) to previous heuristics. *Left:* Histogram of the maximum Q-value computed by *large-value*. *Right:* Histogram of the probability gap computed by *large-prob-gap*. (*all:* all steps in one episode; *attack:* steps that our method attacks)

observe that setting $\lambda$ around the same order of magnitude as

$$\text{Potential reward loss} = \frac{\text{Original return of target agent} - \text{Minimum return of environment}}{\text{Number of steps per episode}}$$

would produce stable results (see Table 4 and Figure 8 in Appendix). This observation could guide the choice of $\lambda$ in different environments.

### 4.4 BEHAVIOR COMPARISON OF THE ATTACK POLICY

To understand to what extent the attack policy learned in our RL framework is different from the heuristics, we attack the target agent by the learned attack policy, and compute the statistics used by the two heuristics at each step. Figure 7 plots the histogram of those statistics in one episode of Pong. We can see that the key steps found by our method spread across different intervals, rather than focusing on the largest intervals. This result suggests that our method does not simply mimic the heuristics, but indeed learns unique key steps to attack. The results of other environments are similar and can be found in Appendix (Figure 9).

## 5 DISCUSSION

We show that the key-step identification problem can be solved by training an attack policy through RL. Compared to existing works, our method learns different key steps without any human knowledge. Results on Atari benchmarks validate our belief that the proposed method may learn more effective key steps. This raises safety concerns on the real-world applications of deep RL agents. One possible future direction is to study how we can improve the RL agent's robustness to this kind of key-step attacks. For example, combining the idea of alternating training (Pinto et al., 2017; Tessler et al., 2019) into our RL framework could be a fruitful research direction. Since the RL agent's robustness is mostly studied in continuous environments (e.g., MuJoCo), we hope that this work attracts more attention into studying this topic in environments with discrete actions.

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

# A  APPENDIX

Table 2: Hyperparameters used during training the attack policy in our RL framework.

| Hyperparameter | Value |
|---|---|
| optimizer | Adam |
| learning rate | 0.0001 |
| batch size | 32 |
| discount factor | 0.99 |
| training steps | 10000000 |
| target network update frequency | 1000 |
| replay buffer size | 100000 |
| learning start step | 20000 |
| learning frequency | 4 |
| exploration type | $\epsilon$-greedy |
| epsilon decay type | linear decay |
| exploration decay horizon | 250000 |
| minimum epsilon during training | 0.01 |
| epsilon during testing | 0.001 |
| adversarial attack method | FGSM |
| maximum norm constraint for perturbation | 0.01 |
| perturbation searching intervals | 100 |

Table 3: Comparison of different $\lambda$ with black-box attacks. The results are averaged over ten episodes with the standard deviation shown in parentheses.

| | | $\lambda$ | | | | | | | | |
|---|---|---|---|---|---|---|---|---|---|---|
| | | $10^1$ | | $10^0$ | | $10^{-1}$ | | $10^{-2}$ | | $10^{-3}$ | |
| Pong | return | 16.3 | (2.1) | 14.7 | (2.2) | -17.2 | (2.5) | -20.2 | (1.0) | -19.7 | (1.6) |
| | attack ratio (%) | 0.0 | (0.0) | 0.1 | (0.0) | 3.1 | (0.4) | 12.4 | (3.5) | 31.5 | (6.9) |
| Space Invaders | return | 787.0 | (457.6) | 571.0 | (294.4) | 539.5 | (309.2) | 422.5 | (267.9) | 269.5 | (183.6) |
| | attack ratio (%) | 6.6 | (1.2) | 9.3 | (2.8) | 46.6 | (4.7) | 73.9 | (7.1) | 57.7 | (6.3) |
| Seaquest | return | 1122.0 | (375.0) | 1038.0 | (463.8) | 1252.0 | (524.2) | 1056.0 | (340.8) | 978.0 | (310.2) |
| | attack ratio (%) | 0.6 | (0.3) | 2.3 | (1.3) | 39.0 | (4.2) | 46.6 | (5.2) | 47.2 | (4.3) |
| Riverraid | return | 9493.0 | (3442.7) | 9891.0 | (2584.1) | 7069.0 | (2488.5) | 9353.0 | (3229.2) | 11176.0 | (2942.4) |
| | attack ratio (%) | 68.0 | (4.6) | 42.8 | (4.0) | 95.7 | (1.9) | 56.2 | (3.1) | 59.9 | (3.5) |

Table 4: Statistics of the target agents averaged over ten episodes.

| | Original return of target agent | Minumum return of environment | Number of steps per episode | Potential reward loss |
|---|---|---|---|---|
| Pong | 18.4 | -21.0 | 2498.9 | $1.58 \times 10^{-2}$ |
| Space Invaders | 1063.0 | 0.0 | 1196.5 | $8.88 \times 10^{-1}$ |
| Seaquest | 1280.0 | 0.0 | 1531.5 | $8.36 \times 10^{-1}$ |
| Riverraid | 12912.0 | 0.0 | 1772.6 | $7.28 \times 10^0$ |

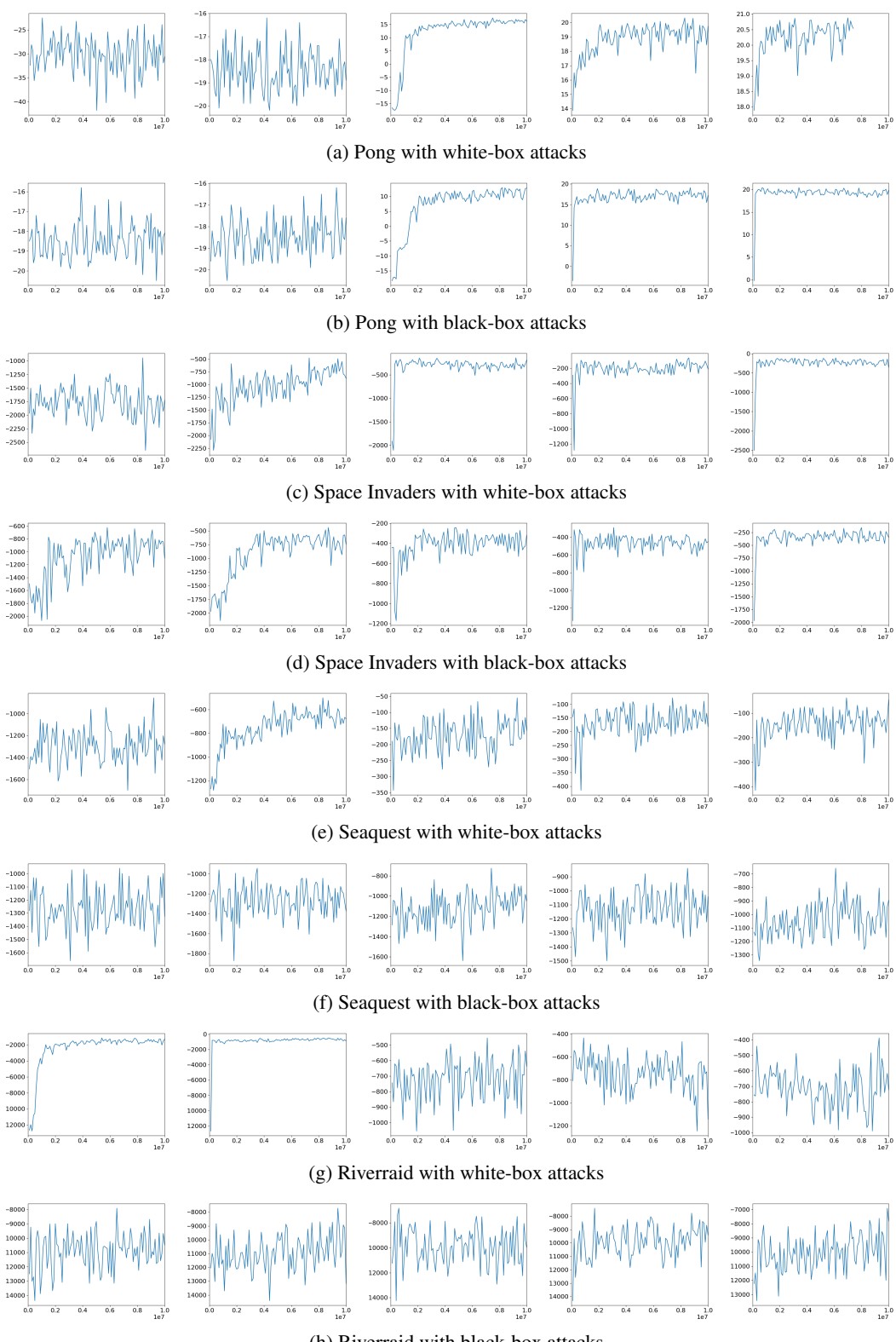

Figure 8: Attack policy's undiscounted return in MDP $M'$ during training ($\lambda = 10^1, 10^0, 10^{-1}, 10^{-2}, 10^{-3}$ from left to right). The attack policy is tested every 100000 steps and we report scores averaged over ten testing episodes.

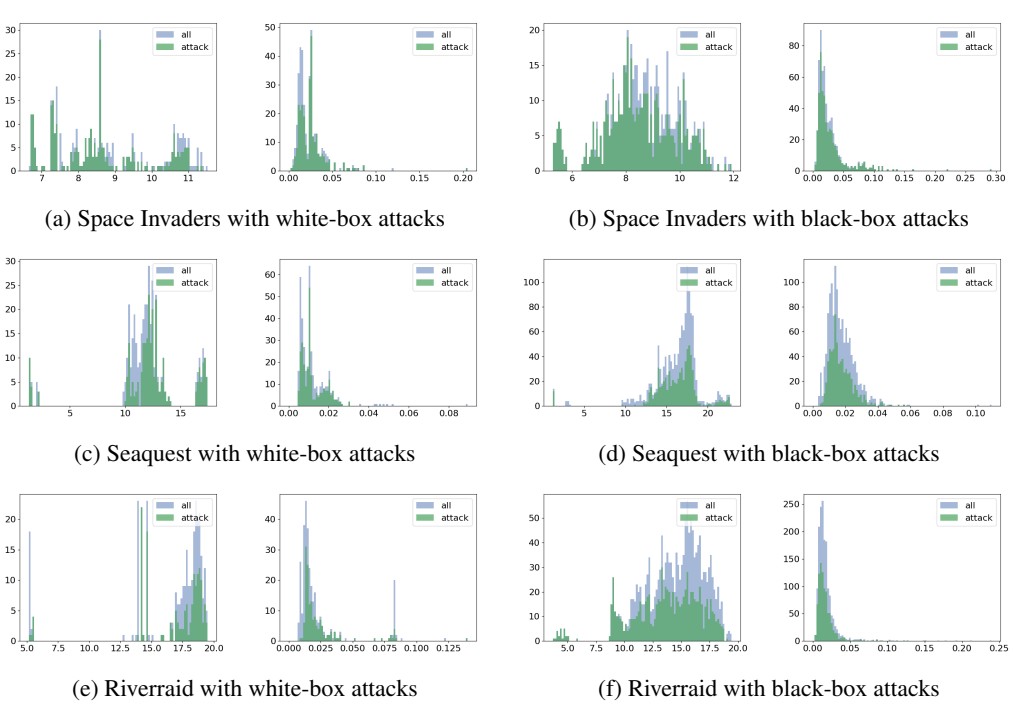

(a) Space Invaders with white-box attacks

(b) Space Invaders with black-box attacks

(c) Seaquest with white-box attacks

(d) Seaquest with black-box attacks

(e) Riverraid with white-box attacks

(f) Riverraid with black-box attacks

Figure 9: Behavior comparison of our method (with $\lambda = 10^{-2}$) to previous heuristics. *Left:* Histogram of the maximum Q-value computed by *large-value*. *Right:* Histogram of the probability gap computed by *large-prob-gap*. (*all:* all steps in one episode; *attack:* steps that our method attacks)

