# OpenReview forum: "Learning Key Steps to Attack Deep Reinforcement Learning Agents"
_ICLR.cc/2020/Conference — Reject_

### Official Review · AnonReviewer3 · 2019-10-18
**Official Blind Review #3**

**Rating:** 3

**Review:**

Contribution:

This work presents a method for performing budgeted attacks on a RL agent where an attacker can apply a perturbation to a limited subset of the observations of the said agent.
To select the "key steps" that needs to be corrupted, they train an opponent RL agent, with a reward crafted such that it should minimize the number of attacks as well as the cumulated reward of the target agent.
Their attack works either in a white-box setting, using FGSM, or in a black-box setting, using a substitute model.

They then proceed to compare to to prior work that use heuristics based on the policy values to select the steps to attack.

The paper is overall well written and easy to follow.

Review:


One major limitation of the work is that the attack rate is not readily modifiable. It uses a penalty term in the reward function with a tunable weight $\lambda$, but changing this weight seemingly requires retraining the opponent from scratch, which is unpractical. By constrast, note that the previous attack methods allow to change the attack rate at will.
More importantly, there is no clear relationship between the value of lambda and the attack rate. With the same lambda, depending on the game, the opponent settles on attack rates varying between 10% (on pong) and 70% (on Riverraid).


The results themselves show only marginal improvement over the baselines, and in the absence of clear error bars / confidence intervals, it is difficult to state the significance of the method. In the particular case of Space Invaders (figure 5b), the proposed method seems tied to the best heuristic (at attack-rate 70%), but the said heuristic reaches the same performance level for much lower attack-rate (as low as 40-50%), implying that the presented method did not do a good job at minimizing the number of attacks.
Overall, a rigorous way of comparing the methods need to be devised. Maybe something like an AUC with proper confidence bounds could do the trick?


In equation (2), you present a target objective function designed using Lagrange relaxation. However, the RL algorithm uses decay (\gamma = 0.99), which means that the resulting function that is effectively minimized is different. Could you clarify the impact of the decay on the lagrange-relaxed objective function?


Could you clarify a bit the section 4.3 on black-box attacks? It seems that you are using a substitute model attack, but it's not clear to me how the substitute is obtained. Is it the same model? How is it trained? Is the attack robust to differences in the algorithm?

Finally, I'm a bit surprised by the choice of DQN as the base algorithm, especially since the chosen framework (Dopamine), offers significantly stronger algorithms (Rainbow or IQN). DQN doesn't even reach perfect score on Pong, which means that the raw policy itself is a bit brittle, since it looses 5 points. Did you try to apply this method on more robust policies?

**Experience Assessment:**

I have read many papers in this area.

**Review Assessment: Checking Correctness Of Derivations And Theory:**

I assessed the sensibility of the derivations and theory.

**Review Assessment: Checking Correctness Of Experiments:**

I assessed the sensibility of the experiments.

**Review Assessment: Thoroughness In Paper Reading:**

I read the paper thoroughly.

---

> ### Author Response · Authors · 2019-11-14
> **Response to Reviewer #3**
>
> We thank Reviewer #3 for the valuable feedback. We have run more experiments and updated a new version. Responses to your comments are listed below:
>
> **Comment**
> One major limitation of the work is that the attack rate is not readily modifiable. [. . .]
> More importantly, there is no clear relationship between the value of lambda and the attack rate. With the same lambda, depending on the game, the opponent settles on attack rates varying between 10% (on pong) and 70% (on Riverraid).
> **Response**
> We agree that our framework does not have direct control on the attack rate, since we replace the hard constraint with a soft penalty parameter. However, we do observe some similar trends in the new experiment results:
> - In the same environment, training with smaller $\lambda$ usually would find attack policies with higher attack rates.
> - Across different environments, setting $\lambda$ around the same order of magnitude as *potential reward loss* (see Section 4.3) tends to produce stable results.
>
> Our results suggest that we should choose $\lambda$ carefully in order to learn effective key steps. We will consider the problem of controlling attack rates as a direction for future improvements.
>
> **Comment**
> The results themselves show only marginal improvement over the baselines, and in the absence of clear error bars / confidence intervals, it is difficult to state the significance of the method. In the particular case of Space Invaders (figure 5b), the proposed method seems tied to the best heuristic (at attack-rate 70%), but the said heuristic reaches the same performance level for much lower attack-rate (as low as 40-50%), [. . .]
> **Response**
> Thank you for the suggestion. We have tested more $\lambda$ and provided the confidence bounds. By setting an appropriate penalty parameter $\lambda$, our method is also able to find attack policy with low attack rate (40-50%) in Space Invaders (see Figure 5b). Although the amount of improvement depends on the environment, we find that
> - Key steps are learnable.
> - Our approach shows comparable performance to competitors across different attack ratios consistently.
> - It is possible to learn more effective key steps using our approach.
>
>
> **Comment**
> In equation (2), you present a target objective function designed using Lagrange relaxation. However, the RL algorithm uses decay (\gamma = 0.99), which means that the resulting function that is effectively minimized is different. Could you clarify the impact of the decay on the lagrange-relaxed objective function?
> **Response**
> Thank you for pointing this out. We agree that the decay changes the minimization problem a little bit. The added decay has similar effects as in the original RL setting. Since the uncertainty of the future may not be fully captured by the current state, the decay could make the attacker emphasize short-term reward loss (of the target agent) more than delayed reward loss. We would consider investigating the decay more in future work.
>
> **Comment**
> Could you clarify a bit the section 4.3 on black-box attacks? [...] Is the attack robust to differences in the algorithm?
> **Response**
> Thank you for the question. We take the substitute agent from pretrained DQN agent in Dopamine. They are trained with the same architecture as the target agent but with different random seeds. We have updated the paper to clarify these details.
>
> I'm a bit unsure about what you mean by “Is the attack robust to differences in the algorithm”. If you mean the robustness of black-box attack in different scenarios, this problem has been studied in [1]. Our setting is the same as "Transferability Across Policies" in their work (Section 5.3.1). While not tested, we believe that our framework is applicable in other black-box settings, since we do not have any assumption on the adversarial attack algorithm.
>
> [1] Sandy Huang, Nicolas Papernot, Ian Goodfellow, Yan Duan, and Pieter Abbeel. Adversarial attacks on neural network policies. 2017.
>
> **Comment**
> Finally, I'm a bit surprised by the choice of DQN as the base algorithm, especially since the chosen framework (Dopamine), offers significantly stronger algorithms (Rainbow or IQN). [. . .] Did you try to apply this method on more robust policies?
> **Response**
> The main reason that we choose DQN is for computation efficiency. Another reason is that we do not want to complicate the perturbation generation procedure. Given that the network structure of Rainbow/IQN is different from traditional classifiers, applying adversarial attacks on them takes additional considerations.
>
> We haven't tried applying to other agents. While evaluating the robustness of stronger agents is indeed an interesting problem, we focus more on the learnability of key steps as an initial work. We'll consider this suggestion in future works.

---

### Official Review · AnonReviewer1 · 2019-10-22
**Official Blind Review #1**

**Rating:** 1

**Review:**


Learning Key Steps to Attack Deep Reinforcement Learning Agents

This paper proposes an extension of existing discrete action image space adversarial attack algorithms.
Instead of choosing the steps by heuristic, the authors propose to choose the key steps by augmenting the reward function with a penalty to decrease the ratio of attacks.

I tend to vote rejection for this paper, given that the proposed algorithms seem incremental compared to the existing algorithms, and the experiments seem not sufficient enough to support the core claim proposed in the paper.

Pros:
- The paper is well written, with sufficient background and related work section for the paper to be self-contained.
- The proposed framework is an interesting and practical framework for attacking RL agents.

Cons:
- The experiment section is insufficient.
More specifically:
1) Results from Figure 5 and Figure 6 seem to disagree with what the authors claim in the paper. In many (the majority) of the environments, the proposed algorithm has only trivial improvement and even worse performance under the same attack rate.

2) It is not very convincing when only one result sample is plotted in Figure 5 and Figure 6.
I think it is necessary to show the performance of the proposed algorithm under different attack rate.
A wide range of candidate penalty parameter lambda should be tested, so that a curve can be fitted for the proposed algorithm similar to the baselines (similar to the one shown in Table 1, but with much more test values).

3) Related to 2), it seems the Lagrange relaxation makes it hard to control the attack rate in the proposed algorithms. How sensitive it is to control the attack rate?

3) Can the authors elaborate on why the algorithms is not too sensitive to the value of penalty in section 4.5?
Table 1, where the performance is almost the same for different penalty parameter, does not necessarily show that the algorithms is not too sensitive to the choice of the penalty parameter.
As mentioned by the author, -21 is the minimum reward (or random reward) an agent can get from Pong.


In general, given the current status of the paper, where there is a lot of room for improvement of experiment section, I will vote for a rejection.


**Experience Assessment:**

I have published in this field for several years.

**Review Assessment: Checking Correctness Of Derivations And Theory:**

I carefully checked the derivations and theory.

**Review Assessment: Checking Correctness Of Experiments:**

I carefully checked the experiments.

**Review Assessment: Thoroughness In Paper Reading:**

I read the paper thoroughly.

---

> ### Author Response · Authors · 2019-11-14
> **Response to Reviewer #1**
>
> We thank Reviewer #1 for the valuable feedback. We respond to your questions below:
>
> **Comment**
> 1) Results from Figure 5 and Figure 6 seem to disagree with what the authors claim in the paper. In many (the majority) of the environments, the proposed algorithm has only trivial improvement and even worse performance under the same attack rate.
> **Response**
> We have performed more experiments, and revised the overly strong claim. Although the amount of improvement depends on the environment, we find that
> - Key steps are learnable.
> - Our approach shows comparable performance to competitors across different attack ratios consistently.
> - It is possible to learn more effective key steps using our approach.
>
>
> **Comment**
> 2) It is not very convincing when only one result sample is plotted in Figure 5 and Figure 6.
> I think it is necessary to show the performance of the proposed algorithm under different attack rate.
> A wide range of candidate penalty parameter lambda should be tested, so that a curve can be fitted for the proposed algorithm similar to the baselines (similar to the one shown in Table 1, but with much more test values).
> **Response**
> Thanks you for the suggestion. We have tested more $\lambda$ and updated the paper. Figure 5 and Figure 6 plot the mean score with one standard deviation across different attack rates.
>
>
> **Comment**
> 3) Related to 2), it seems the Lagrange relaxation makes it hard to control the attack rate in the proposed algorithms. How sensitive it is to control the attack rate?
> **Response**
> Since we replace the hard constraint with a soft penalty parameter, how to control the attack rate is indeed a tricky problem. Empirically, we observe that training with smaller $\lambda$ tends to produce attack policies with higher attack rates (see Table 1 and Table 3). However, we do not have direct control of the attack rate in the current framework. We will consider this problem as a direction for future improvements.
>
>
> **Comment**
> 3) Can the authors elaborate on why the algorithms is not too sensitive to the value of penalty in section 4.5? [. . .]
> **Response**
> Originally, by "not too sensitive to $\lambda$" we mean that our framework is able to learn effective key steps for a number of different $\lambda$. To avoid confusion, we removed this statement in the updated version. Also, based on the new experiments, we add more analysis on how to choose an appropriate $\lambda$. In particular, we observe that setting $\lambda$ around the same order of magnitude as *potential reward loss* (see Section 4.3) tends to produce stable results.

---

### Official Review · AnonReviewer2 · 2019-10-24
**Official Blind Review #2**

**Rating:** 3

**Review:**

This paper proposes to learn the ‘key-steps’ at which to to apply an adversarial attack on a reinforcement learning agent. The framing of this problem is a Lagrangian relaxation of a constrained minimization problem which takes the form of an RL problem itself, where the attacking agent’s reward is the negative reward of the target agents plus a penalty (lambda, hyper-parameter) for choosing to attack. The attacking agent’s action space is binary, attack or no attack.

The RL approach is compared with a random attack policy and two heuristic methods for attacking agents in games on the Atari benchmark.

The setting addressed in this work, where the attacker only learns whether/when to attack or not is a greatly simplified version of the full problem. It is (in my opinion, but feel free to correct me) likely that the type of perturbation also being learned has even greater potential, but is (obviously) much harder.

Additionally, although the authors mention this as future work, I think the co-training setting is particularly interesting. As the authors suggest, this could lead to more robust target agents, but additionally I wonder if the type and difficulty of attacks would vary as the target agent trains.

Besides the (picky) complaints, I think the problem formulation is quite reasonable. Again, I find the larger problem extremely interesting, but perhaps this is far too intractable right now. The formulation is straightforward, so while I recognize it as a contribution, it is not a particularly large one.

On the other hand, the experimental results appear somewhat weak to me.

If you vary the lambda parameter (as in Table 1, but for all games) you should be able to get similar line plots in Figure 5&6 for the RL approach, which would give a much better comparison for the trade-offs as you get for the heuristic methods. I think this comparison would be very interesting and strengthen the existing results in those figures.

The results, currently, do not appear very significant because (1) the gap between the RL solution and the heuristics is very small and (2) these *appear* to be single runs without standard deviations displayed. Can you argue for why these results *are* in fact significant (statistically or otherwise)?

Another question raised by the results is how performance of the attacking policy varies with training. The authors point out that the training is quite small compared to the target policy, but is that because it has already found the best solution it can in that time? How would it improve with more training?

Small note on Figure 7, I think the point for these would be better made by normalizing the respective histograms.

Update:

Thank you for your responses and updating with new results. I think these provide a much better picture of the performance of the RL-based system. Although I am revising my score upward, I still think this is generally a rejection. Obviously I still think the full problem is interesting, but even the key step identification problem would be publishable if the performance was improved or further analysis helped me understand why RL is not doing better (since the heuristics are after all just heuristics). Good luck on future versions of this work.

**Experience Assessment:**

I have read many papers in this area.

**Review Assessment: Checking Correctness Of Derivations And Theory:**

N/A

**Review Assessment: Checking Correctness Of Experiments:**

I carefully checked the experiments.

**Review Assessment: Thoroughness In Paper Reading:**

I read the paper at least twice and used my best judgement in assessing the paper.

---

> ### Author Response · Authors · 2019-11-14
> **Response to Reviewer #2**
>
> We thank Reviewer #2 for the valuable suggestions. We have updated a version based on your feedback. Responses to your comments are listed below:
>
> **Comment**
> The setting addressed in this work, where the attacker only learns whether/when to attack or not is a greatly simplified version of the full problem. [. . .] but additionally I wonder if the type and difficulty of attacks would vary as the target agent trains.
> **Response**
> We agree that learning the type of perturbation is an interesting direction and may lead to stronger attacks. Our work focuses more on the learnability of key steps, which is unexplored in previous studies. In order to have fair comparisons against previous methods, we fix the perturbation generation part as a subroutine. We would consider further generalizations in the future.
>
>
> **Comment**
> If you vary the lambda parameter (as in Table 1, but for all games) you should be able to get similar line plots in Figure 5&6 for the RL approach, [. . .] The results, currently, do not appear very significant because (1) the gap between the RL solution and the heuristics is very small and (2) these appear to be single runs without standard deviations displayed. Can you argue for why these results are in fact significant (statistically or otherwise)?
> **Response**
> Thank you for the suggestion. We have run more experiments with different $\lambda$, and updated the results in Section 4.2 and 4.3. Figure 5 and Figure 6 plot the mean score with one standard deviation.
>
> Out method do not incorporate human knowledge, but is able to perform comparably to heuristics across different attack ratios consistently, and achieves superior performances in some environments. For example, we observe significant improvements in Pong. These results demonstrate that the attacker trained by our framework learns effective key steps and has the potential to outperform human-designed heuristics.
>
>
> **Comment**
> Another question raised by the results is how performance of the attacking policy varies with training. The authors point out that the training is quite small compared to the target policy, but is that because it has already found the best solution it can in that time? How would it improve with more training?
> **Response**
> We choose a rather small training budget due to limited computation resources. Empirically, we observe that if $\lambda$ is set in an appropriate range, the attack policy often converges in 10M training steps. We have provided more details during training in Appendix (Figure 8). We do not exclude the possibility that the attack policy might improve if given more training steps.

---

### Author Response · Authors · 2019-11-14
**Response to all reviewers**

We thank all reviewers for the valuable and constructive feedback. We have updated the paper based on the reviewers' comments and suggestions. The updates are summarized as follows:

- We ran more experiments. We train the attack policy using five different values of $\lambda$ across all environments with both white-box and black-box attacks.
- We rewrote the experiment part. Section 4.2 compares the performance between our method and competitors. Section 4.3 investigates the effect of penalty parameter $\lambda$.
- Based on the new results, we revised some overly strong claims and provided more analyses.

The important updates are highlighted in red. Any further suggestion or feedback is welcomed.

**Updates at Nov. 15 12:58 (Pacific Time)**
We have uploaded the final version (without red highlights).

---

### Decision · Program_Chairs · 2019-12-19

**Decision:**

Reject

**Comment:**

This paper considers adversarial attacks in deep reinforcement learning, and specifically focuses on the problem of identifying key steps to attack. The paper poses learning these key steps as an RL problem with a cost for the attacker choosing to attack.

The reviewers agreed that this was an interesting problem setup, and the ability to learn these attacks without heuristics is promising. The main concern, which was felt was not adequately addressed in the rebuttals, was that the results need to be more than just competitive with heuristic approaches.

The fact that the attack ratio cannot be reliably changed, even with varying $\lambda$ still presents a major hurdle in the evaluation of the proposed method.

For the aforementioned reasons, I recommend rejecting this paper.